# Socio-economic factors and its influence on the association between temperature and dengue incidence in 61 Provinces of the Philippines, 2010–2019

**Xerxes Seposo** [1,2,3]*, **Sary Valenzuela**[3], **Geminn Louis Apostol**[3]

**1** Department of Hygiene, Hokkaido University, Sapporo, Hokkaido Japan, **2** School of Tropical Medicine and Global Health, Nagasaki University, Nagasaki, Japan, **3** Ateneo School of Medicine and Public Health, Ateneo de Manila University, Pasig, Philippines

* seposo.xerxestesoro@pop.med.hokudai.ac.jp

## Abstract

### Background

Temperature has a significant impact on dengue incidence, however, changes on the temperature-dengue relationship across axes of socio-economic vulnerability is not well described. This study sought to determine the association between dengue and temperature in multiple locations in the Philippines and explore the effect modification by socio-economic factors.

### Method

Nationwide dengue cases per province from 2010 to 2019 and data on temperature were obtained from the Philippines' Department of Health–Epidemiological Bureau and ERA5-land, respectively. A generalized additive mixed model (GAMM) with a distributed lag non-linear model was utilized to examine the association between temperature and dengue incidence. We further implemented an interaction analysis in determining how socio-economic factors modify the association. All analyses were implemented using R programming.

### Results

Nationwide temperature-dengue risk function was noted to depict an inverted U-shaped pattern. Dengue risk increased linearly alongside increasing mean temperature from 15.8 degrees Celsius and peaking at 27.5 degrees Celsius before declining. However, province-specific analyses revealed significant heterogeneity. Socio-economic factors had varying impact on the temperature-dengue association. Provinces with high population density, less people in urban areas with larger household size, high poverty incidence, higher health spending per capita, and in lower latitudes were noted to exhibit statistically higher dengue risk compared to their counterparts at the upper temperature range.

**Data Availability Statement:** The data set for dengue incidence was acquired with permission from the Philippines Department of Health Epidemiological Bureau, while that of the provincial

mean weekly temperatures were acquired from the Philippine Atmospheric, Geophysical and Astronomical Services Administration. The data sets may be accessed upon formal request by contacting the data source owners directly at nec.doh@gmail.com +63 (02) 8651 7800 and (+632) 8927-1541 or information@pagasa.dost.gov.ph, respectively. R codes to perform the analyses are available upon formal request from the Department of Hygiene, Graduate School of Medicine, Hokkaido University at eisei@med.hokudai.ac.jp.

**Funding:** XTS was supported by the Japan Society for the Promotion of Science Project number 22K17374. XTS and GLA were supported by Ateneo de Manila University URC 2022-18. The funders had no role in study design, data collection and analysis, decision to publish, or preparation of the manuscript.

**Competing interests:** The authors have declared that no competing interests exist.

## Conclusions

This observational study found that temperature was associated with dengue incidence, and that this association is more apparent in locations with high population density, less people in urban areas with larger household size, high poverty incidence, higher health spending per capita, and in lower latitudes. Differences with socio-economic conditions is linked with dengue risk. This highlights the need to develop interventions tailor-fit to local conditions.

## Author summary

This study examined the effect of temperature on dengue in the 61 Provinces of the Philippines. We found that as temperature increases, so does the risk of dengue. However, the increase in the risk of dengue plateaus at 27.5 degrees Celsius, and beyond that temperature the risk decreases. The association between temperature and dengue incidence at high temperatures was stronger among provinces with high population density, less people in urban areas with larger household size, high poverty incidence, higher health spending per capita, and in lower latitudes compared to their counterparts. In particular, results showed that people in the rural areas had higher dengue risk than those in urban areas, which is contrary to expectations that dengue is an urban-dwelling related disease. On another hand, we found that high health spending per capita does not equate to better health outcomes. Efficiency of use of resources may come into play in terms of improving the health outcomes and not just the quantity or amount of the difference in the dengue risk by socio-economic factors highlights the need for a tailor-fit dengue response.

## Introduction

Dengue, transmitted by the mosquito vector *Aedes aegypti*, remains a major public health concern in tropical areas [1]. Over the last 50 years, dengue incidence has increased globally by 30-fold and 70% of the global burden of disease comes from Asia [2]. In 2019, the Philippines declared dengue a national epidemic of concern with 420,453 cases. Similar increases in dengue morbidity have been noted in Vietnam and Malaysia [3]. Heightened dengue transmission may be partly driven by the increase in international travel, unplanned urbanization, lack of effective vector management, climate change, and poor socio-economic status, among others [1,4].

Literature identifies ambient temperature as an important meteorological risk factor for dengue transmission [5]. A systematic review by Li et. al. [5] revealed that the minimum temperature of 18 degrees Celsius is a critical limiting risk factor for dengue transmission, with high ambient temperatures promoting vector proliferation. Other global meta-analyses pose that dengue epidemics have a critical range of 22–29 degrees Celsius and that extremely higher temperatures reduce vector survival time [1,6]. In the advent of climate change, several studies have shown that current projections of 2 degrees Celsius by the end of the next century will result in an expansion of latitude and altitude range and duration of transmission of dengue [5,7,8] or even a shift in geographical regions affected by vector-borne disease outbreaks [9].

There have been similar climate change-dengue studies conducted in Southeast Asia, observing an increase of dengue risk within a specified range of high temperatures. In

Thailand, vector efficiency increased in temperatures from 20–30 degrees Celsius, affecting annual cyclic patterns of dengue hemorrhagic fever epidemics in Bangkok [10]. Similar results were noted with the studies in Singapore [11–13]. A study in China [14] observed an inverse-U shape association between temperature and dengue. One study in the Philippines observed significant interaction effects between seasons, temperature, and altitude on the hatch rates (HRs) of *A. aegypti* larvae and reproductive outputs (ROs) of mosquitoes [15]. HRs in both seasons were highest at 25 degrees Celsius and lowest at 38 degrees Celsius. ROs were highest at 25 degrees Celsius in the wet season and at 18 degrees Celsius in the dry season. Further-more, the study noted that the incidence of the dengue-causing mosquito phenotype was found to be highest at 18 degrees Celsius and present even at 38 degrees Celsius in both sea-sons, suggesting larval adaptation to global warming [15].

Several temperature-dengue studies have documented evident heterogeneity in the associa-tion, which may potentially be due to socio-demographic and economic factors innate to the specific locations [16–19]. Certain vulnerable sub-populations have been observed to have higher infectious disease risk incidence such as those with higher poverty incidence, urban to population ratios, and smaller government health expenditures, as they have lower adaptive capacity and are more susceptible to negative health outcomes when exposed to environmental hazards [20–22]. While noted, examination of the contributors to heterogeneity of the linear association between temperature and dengue have been relatively scarce, with little to no evi-dence on how these socio-demographic and economic factors impact the non-linear tempera-ture-dengue association. Against this backdrop, this study sought to determine the association between temperature and dengue in multiple locations in the Philippines, focusing on the role of socio-demographic and economic factors in the temperature-dengue association.

## Methods

### Ethics statement

The Ateneo de Manila University Research Ethics Committee waived the need for ethical approval and the need to obtain consent for collection, analysis, and publication of secondar-ily, retrospectively obtained for this non-interventional study. All methods were performed in accordance with the relevant guidelines and regulations.

### Data source

Nationwide, weekly notifiable dengue cases per province (n = 81) from 2010–2019 were obtained from the Philippines' Department of Health–Epidemiological Bureau. In the Philip-pines, case surveillance is coursed through the Philippine Integrated Disease Surveillance and Response, which is subsequently managed by the Philippine Department of Health's Epidemi-ological Bureau. In this study, provinces (n = 61) with more than 300,000 population were uti-lized for analysis. Provinces with less than 300,000 population had a higher percentage of 0 observations, affecting statistical power and the subsequent uncertainty in the estimation, and were thus excluded from the current study.

In the absence of consistent background monitoring data for each of the provinces, hourly temperature was obtained from ERA5-land, which has a spatial resolution of 9km on a reduced Gaussian grid. Developed by the Copernicus Climate Change Service (C3S) on behalf of the European Union, ERA5-land is a publicly available enhanced global dataset that has shown accurate temperature readings in Europe and Central Asia [23–25]. Hourly tempera-tures were then aggregated to daily 24 hour mean temperature, and thereafter was aggregated into a weekly measure. Statistical correlation of temperature obtained from ERA5-land and those locations with existing background monitoring stations is high. For the case of Manila

City, Pearson's correlation coefficient was at 0.88 (p-val < 0.05), whereas for Cebu City it was at 0.86 (p-val < 0.05) (S1 Fig). Over the same period, daily temperature was aggregated by epidemiological week (EpiWeek) to match the dengue incidence data. On a similar note, several studies have utilized ERA5-land in examining the effect of temperature on specific health outcomes [26–28].

Potential effect modifiers were obtained from literature as shown in S1 Table. The socioeconomic factors included were poverty incidence, population density, people living in urban areas, number of people per household, health expenditure, and latitude per province, which were sourced from relevant literatures [20–22,29–34]. The socio-economic factors used were province specific. All were originally yearly data, except for latitude. These terms have been operationalized as shown in S2 Table.

## Statistical analyses

We employed a generalized additive mixed model (GAMM) in examining the nationally representative temperature dengue association. Dengue data was assumed to follow a negative binomial distribution. Subsequently, we included a crossbasis term for temperature and adjusted for several covariates as shown in Eq 1.

$$Y_{t,i} \sim Negative\ binomial \tag{Eq1}$$

$$Y_{t,i} \sim \alpha + logpop + cbtemp + s(Epiweek, k = 4) + (1|Year) + (1|Province) + Y_{t-1,i} + \varepsilon$$

$Y_{t,i}$ is the weekly dengue incidence in time ($t$) of province ($i$). The weekly dengue incidence is assumed to follow a negative binomial distribution, which is similar to previous studies [35]. $\alpha$ is the intercept. *logpop* is the log of the population, which is an offset of the changing population sizes [5]. *cbtemp* is a crossbasis function of the exposure and lag dimensions of temperature, with both dimensions parameterized with 4 degrees of freedom (df) using natural cubic splines, respectively. The maximum lag was set at 18 weeks (or 4.5 months), based on the maximum lag sensitivity analyses (S2 Fig) and loosely similar to other studies whose maximum lag was set around 20 weeks [13,36]. After further examination, a seasonal pattern can be noted in Fig 1, with a peak of the dengue cases around 32nd/33rd week. Thus, we adjusted for the seasonality of dengue cases through a 4 df, based on a sensitivity analysis in S3 Fig. Owing to the variations in the annual distribution of cases per year (S4 Fig), we added a random effect term of year; (1|Year). A random effect term of Province (1|Province) was also added to emulate a (partial) pooling effect in estimating the nationally representative association. Several studies have noted the potential immunity in the population, which could inflate the susceptible population in the subsequent time scale. To model this phenomenon, Imai et al. [37] utilized the previous time scale's case as a potential adjustment. Here, we adjusted for the previous week's case; $Y_{t-1,i}$. $\varepsilon$ is the error term.

Effect modification by socioeconomic factors was also examined through an interaction analysis. In brief, the 5th and 95th percentile values for the specific effect modifier represented the low and high percentiles. The respective values of the effect modifier were then deducted from the 5th or the 95th percentile values, which effectively centers these values around the low and high percentiles. The temperature crossbasis term is then multiplied by the centered effect modifier in either (low or high) percentiles. The resultant crossbasis represents the exposure-lag-response dependency corresponding to the percentile centering point of the effect

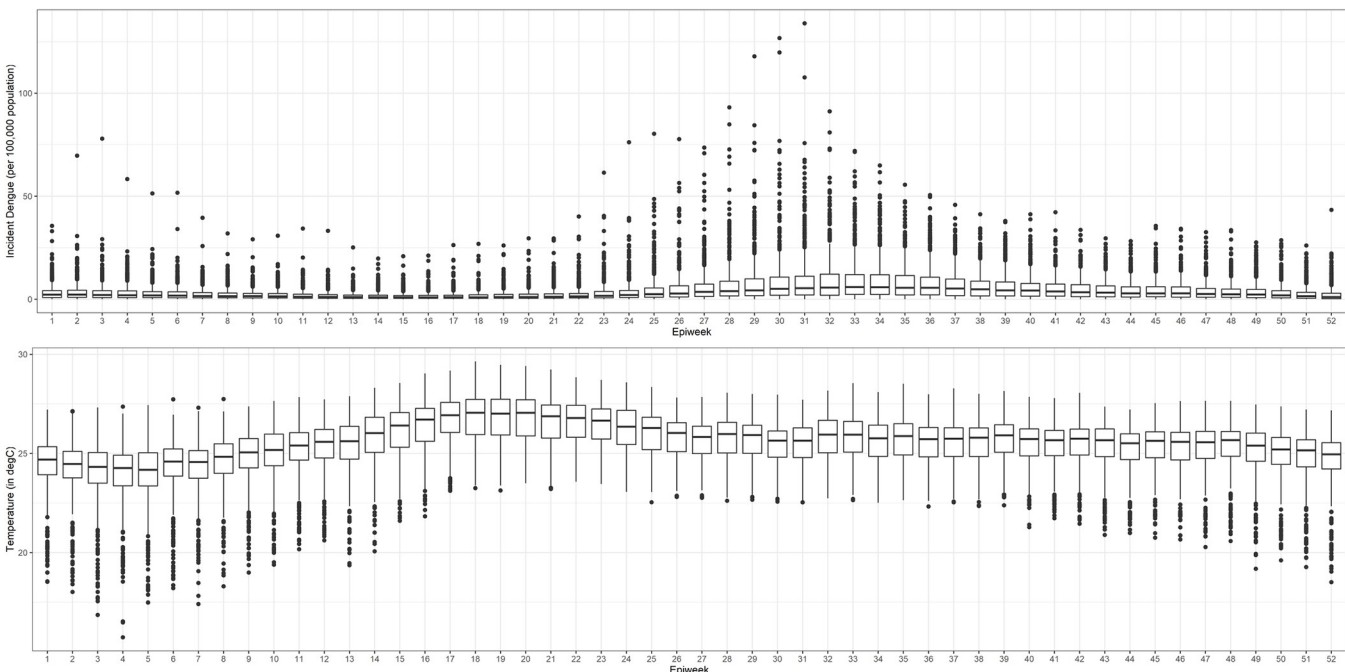

**Fig 1.** Boxplot diagrams of average dengue incidence (upper panel) and average recorded temperatures (lower panel) occurring per week in a year in the Philippines.

modifier [38].

$$Y_{t,i} \sim \alpha + logpop + cbtemp + effmod_j + cbtemp : effmod_j + s(Epiweek, k$$
$$= 4) + (1|Year) + (1|Province) + Y_{t-1,i} + \varepsilon \qquad (Eq2)$$

Here, $effmod_j$ represents the respective effect modifier centered at either percentile $j$ (5th or 95th). All analyses were implemented using R statistical programming version 4.1.3 through the following packages: "dlnm" [39], "mixmeta" [40], "mgcv" [41], "lubridate" [42], and "dplyr" [43].

## Results

### Descriptive analyses

A mean, weekly dengue incidence of 2.128 was recorded from 2010–2019 across 61 provinces in the Philippines, with a range from 0 to 133.925 per EpiWeek. Recorded temperatures each week in each province had a median of 25.62 and a range of 15.73–29.64 per EpiWeek. The population per province varied greatly from 29,2781–13,484,462 people and a median of 89,0451 (Table 1) per EpiWeek.

Fig 2A shows the distribution of dengue incidence per 100,000 population from 2010–2019, in all study provinces across the three major island groups of Luzon (north), Visayas (middle), and Mindanao (south). The top three provinces recording the highest dengue incidence were the highly urbanized provinces of Davao del Sur (11.14 per 100,000 population), South Cotabato (8.19 per 100,000 population), and Misamis Oriental (6.61 per 100,000 population). Provinces reporting the lowest dengue incidence were Lanao del Sur (0.29 per 100,000 population), Tawi-tawi (0.26 per 100,000 population, and Sulu (0.10 per 100,000 population), which are high-conflict areas in Mindanao [44,45]. Fig 2B indicates the average reported temperature per

**Table 1.  Descriptive summary statistics for dengue cases, temperature, population, and dengue incidence across the 61 Provinces in the Philippines from 2010–2019.**

|  | Median | Percentiles | | | | Interquartile Range (IQR) | Range |
|---|---|---|---|---|---|---|---|
|  |  | 1st | 25th | 75th | 99th |  |  |
| Dengue Cases | 23 | 0 | 7 | 64 | 577.81 | 57 | 0–2,748 |
| Temperature (°C) | 25.62 | 20.92 | 24.64 | 26.45 | 28.23 | 1.78 | 15.73–29.64 |
| Population | 89,0451 | 308,985 | 617,333 | 1,697,050 | 12,877,253 | 1,079,717 | 29,2781–13,484,462 |
| Dengue Incidence | 2.128 | 0 | 0.787 | 5.290 | 31.180 | 4.503 | 0–133.925 |

province in the same time period. The provinces that reported the highest temperatures between 26–27 degrees Celsius clustered around central and southern Luzon, Visayas, and the northern areas of Mindanao. The highest recorded daily mean temperatures were in Tawi-tawi (27.04 degrees Celsius), Sulu (26.88 degrees Celsius), and Basilan (26.67 degrees Celsius).

Seasonal trend in dengue incidence is apparent (as shown in Fig 1, Upper Panel) with the gradual increase starting in EpiWeek 19 (1.05 per 100,000 population), peaks in EpiWeek 36 (5.56 per 100,000 population), then steadily decreases until the end of the year. In the lower panel, the range of the median value of the average temperature is around 23 to 27 degrees Celsius, peaking around EpiWeek 20 (26.78 degrees Celsius).

## Association between temperature and dengue incidence

Fig 3 depicts the overall exposure-response association. Dengue-related risk is linearly increasing from the minimum temperature of 15.8 degrees Celsius and peaking at 27.5 degrees Celsius before declining. The inverted U-shaped risk pattern between temperature and dengue remains to be robust even after adjusting for precipitation as shown in S5 Fig. On the other hand, cumulative lag-response association at the 90th percentile, relative to the 50th percentile of the prediction interval, shows a reduced risk at the earlier lags but with a gradual increase in latter lags peaking

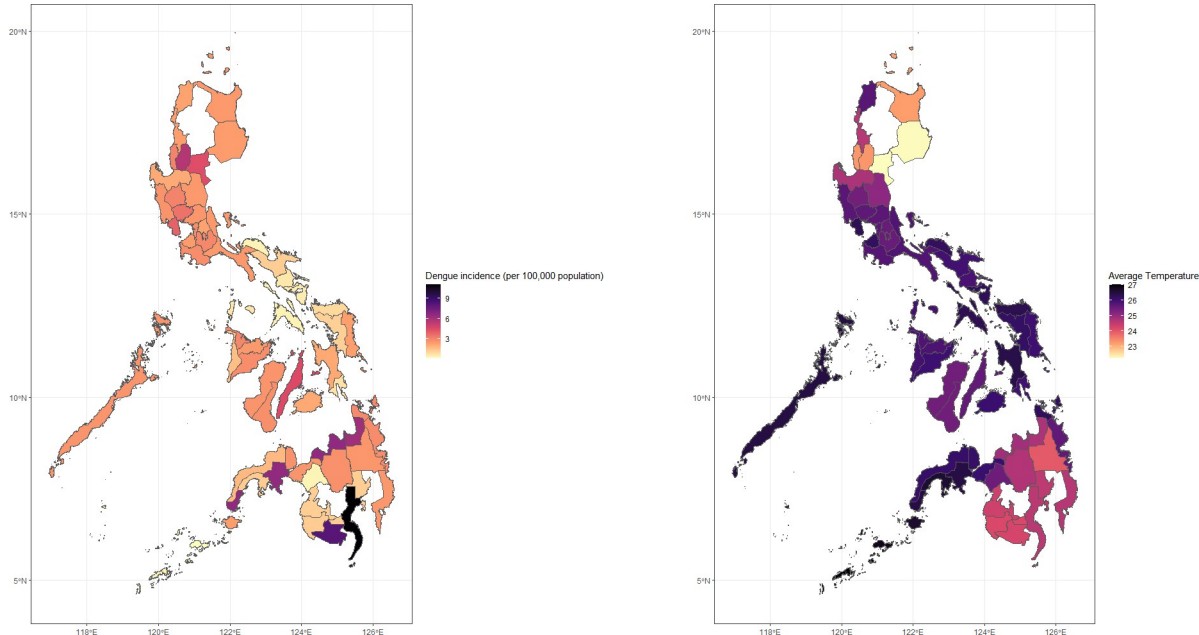

**Fig 2.** Heat map* of dengue incidence cases (left) and average temperature (right) per region in the Philippines from 2010–2019. * **Fig 2 is created using R package "ggmap".**

**Overall cumulative temperature−dengue association**

**Fig 3. Nationally representative exposure-response function of increasing temperature with dengue incidence.**

at around 16 weeks (or approximately 4 months), followed by a subsequent reduction in the risks (Fig 4). Further examination of the exposure- and lag-response associations by specific lag period (Weeks 0, 3, 6, 9, 12 and 18) and temperature percentiles (1st, 5th, 10th, 90th, 95th, and 99th), revealed nearly consistent patterns with the overall exposure and lag associations (in S6 Fig).

### Effect modification by socio-economic factors

Socio-economic factors had varying impact on the temperature-dengue association (in Fig 5). Operationalization of the low and high for each socioeconomic variable is relative to each socioeconomic variable's 5th and 95th percentiles, respectively. Provinces with high population density, less people in urban areas with larger household size, high poverty incidence, higher health spending per capita, and in lower latitudes were noted to exhibit statistically higher dengue risk compared to their counterparts at the upper temperature range (Figs 5 and S7). Modifications in the lower temperature range were also observed for all effect modifiers, except for latitude. Main effects of the effect modifiers are shown in S3 Table.

### Discussion

Nationally representative exposure-response function between temperature and dengue in the Philippines was noted to have an inverse U-shaped, non-linear association, with the relative

**Lag−response association at 90th temperature percentile**

**Fig 4. Nationally representative lagged temperature-dengue association at the 90th temperature percentile relative to the 50th percentile of the prediction interval.**

risk peaking at 27.5 degrees Celsius. In Guangzhou, China, both Wu et al. [14] and Xiang et al. [46], also noted an inverse U-shaped temperature-dengue association. The decline in the risk beyond 27.5 degrees Celsius can be partially explained by the physiological mechanisms affecting vector survival. Extremely hot temperatures beyond maximum temperatures have an inverse effect on fecundity, blood size meal, and vector viability, reducing mosquito survival time [6,47]. Dengue fever risk was seen to increase after extremely high temperatures due to accelerated incubation and increased mosquito population, although dengue lifespan and egg survival rate was decreased [48]. Hawley [49] observed that the optimal thermal conditions for *A. albopictus* survival is around 20 to 27 degrees Celsius. Another study by Servadio et. al. [9,13], presented similar results of a similar parabolic curve, with maximum development rate in vector-borne diseases at temperatures between 28 and 32˚C and none under 18C or over 34˚C in South and Southeast Asian countries. This is further supported by other studies [50,51], which observed a decline in vector's survival rate beyond 28 degrees Celsius.

Our results have shown a 16-week (or approximately 4 months) lag association between temperature and dengue incidence as shown in Fig 4. Previous studies have also utilized longer lag periods to examine the lag association. In Singapore, Hii et al. [13] utilized a 20-week maximum lag to examine the effects of weather variables (including temperature) on dengue incidence. Results showed higher relative risks of dengue cases at a time lag of 3–4 months, which

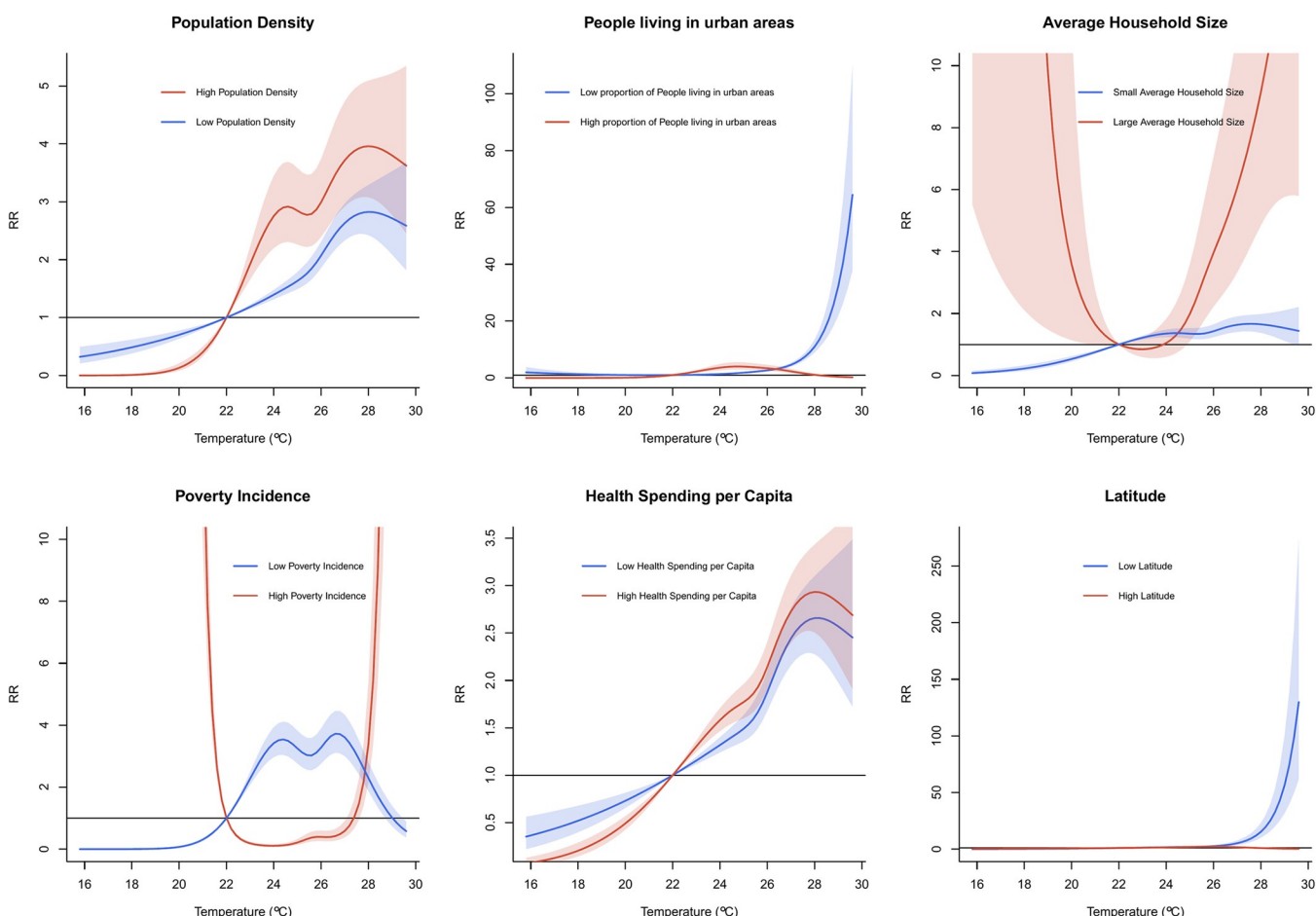

**Fig 5. Effect modification of the temperature-dengue association by socio-economic variables.**

is consistent with our study findings. Similarly, Hii et al. [36] also utilized a 20-week maximum lag in examining the optimal lead time for dengue forecasting (parameterized by weather variables), with an optimal lead time at around 3 months. Koh et al. [52] also found that dengue incidence in 2005 was highly associated with weekly mean temperature at a lag time of 18 weeks. Heng et al. [53] noted that dengue incidence occurred at a lag time of 8–20 weeks following elevated temperature. Similar 3-month lag associations have been observed in Bangkok [54] and Taiwan [14]. The delayed association between temperature and dengue incidence may potentially be related to the incubation periods in host–vector–pathogen transmissions cycle plus reproduction, maturation and survival rate of vector [13].

Similar to previous studies, higher population density was associated with a visually higher temperature-dengueran association than low population dense areas [33,55]. In Indonesia, Nuranisa, et al. [56] noted a positive correlational association between dengue incidence and population density. These highly dense areas provide suitable conditions for vector proliferation and subsequent disease transmission [57]. Noteworthily, urbanization revealed a reduced risk for temperature-related dengue incidence, contrary to prior literature, which associates them as well-documented risk factors [17,19,54,58–61]. Unplanned urbanization promotes close proximity to larval habitats, dengue-prone water storage practices, and rapid population movement [1,59]. Notably, the top three provinces for dengue incidence, Davao del Sur, South

Cotabato, and Misamis Oriental, are some of the more urbanized cities and more populated areas in the country [45]. These results suggest that the connection between dengue, human behavior, and temporal relationships is more complex than previously thought. As to whether dengue remains primarily a rural disease as a result of increased rural transmission or due to diminishing incidence rates in urban areas due to concentrated vector control efforts and increased health spending in urban areas, remains to be elucidated. Dengue transmission may indeed be increasing in rural areas due to the interplay of additional modifying variables that were not included in the study due to lack of geographical and behavioral data, such as forest cover or the presence of vegetation, open water containers, and degree of intrahousehold crowding, which are known dengue risk factors [1,17,19,54,58,59,61,62]. One study in Vietnam [34] found that risk of dengue was higher in rural areas due to the lack of piped water supply. Another study in Rio De Janeiro noted that while highly-populated, highly-urbanized areas displayed increased dengue incidence, so did other areas with low human density but with medium to highly-vegetated neighborhoods [62]. Further investigations are required to prove these hypotheses. While our analysis provides hindsight on the role of province-specific characteristics and its influence on the relationship between dengue and temperature, we cannot ascertain the possible mechanisms that are related to the observed associations.

Average household size was found to have higher temperature-related dengue risk than small households. Similar results were seen in New Caledonia [16]. Its known vectors, *Aedes aegypti* or *Aedes albopictus*, exhibit limited flight capacity and remain in close proximity to their breeding water containers [60,63]. Domestic workers and housewives are also more vulnerable to dengue compared to those who spend less time at home [17,59]. Larger families may also have larger dwellings, which may have more water-storage containers, such as flowerpots, vases, and lush areas that can serve as breeding sites [60]. As dengue cases have also been observed to cluster within households and other countries [63–65], intrahousehold dengue may be more prevalent than interhousehold dengue transmission. In contrast, a study in Brazil found large household sizes were protective against dengue risk, as smaller families (<7 people) were found to be living in more cramped spaces than their larger counterparts in the studied area [17]. This is in concurrence to the results observed by Zambrano, et. al. [59] who argued that the increased ratio of household members to the number of rooms, otherwise known as "crowding", was more predictive of dengue incidence than the absolute number of household members. Risks apparent in the lower temperature range among large household size families may potentially also be related to the persistence of disease transmission whose range falls between 21.6–32.9 degrees Celsius [46] coupled by the crowding of the household members.

Poverty incidence was also found to modify the temperature-dengue association in both low and high temperature ranges. Several studies associate poverty with dengue risk due to substandard housing, solid waste mismanagement, and sewerage systems that favor close proximity to larval habitats [17,58,59]. Mulligan et. al. [61] argues that dengue's epidemiological and transmission characteristics in relation to poverty should be interpreted in the context of how these specific poverty-related factors (i.e. household overcrowding, absence of air conditioning, low education, low income, poor housing quality, self-assessment of economic situation as poor, low family income, higher population density, and inappropriate use of covers on water containers, among others) affect the biology and ecology of the vector *Aedes aegypti*. In addition, poverty incidence may simply represent a variety of behavioral and exposure differences of the population. Resource-poor households/populations tend to work mostly during extreme conditions, including that during low temperatures and high temperatures in contrast to those resource-rich households/populations. Populations in resource-poor locations would prioritize more food and other basic necessities than personal risk prevention of dengue (e.g.

mosquito repelling lotion, long sleeves, etc.). Since dengue transmission may persist in a range of 21.6–32.9 degrees Celsius [46]. If we chart against Fig 4 (poverty incidence panel), we note that the range 21 to 29 degrees Celsius, which is well within the range of dengue transmission, hence the potential high risk at these extremes for the high poverty incidence populations. The low risk at the middle may also be similarly related to low exposure of the population since these are temperatures that are conducive for most less labor-intensive jobs, whereby these populations don't engage with. These are mostly assumptions that remain to be elucidated in future studies.

Contrary to the expected outcome that provinces with higher health expenditures would exhibit a reduced risk of temperature-related dengue risk due to the potential focus of these resources on overall health enhancement, our findings indicate a higher temperature-related dengue risk in areas with greater health spending as opposed to those with lower health expenditures The increased dengue risk amidst high health spending may seem counterintuitive, however, health spending does not necessarily equate to improved health. Dieleman et. al. [66] conducted a study analyzing factors associated with increases in US health care spending from 1996 to 2013. The study found that despite the substantial increase in health care spending during this period, there was no significant improvement in health outcomes, such as mortality rates or disease burden. The linkage between health care spending and improvement of health outcome is not direct and may be mediated by several factors. In Sub-Saharan Africa (SSA), Makuta and O'Hare [67] examined the relationship between public spending on health, quality of governance, and health outcomes. Their findings indicated that health spending has a statistically significant impact on improving health outcomes, such as under-five mortality and life expectancy. However, the impact of health spending on health outcomes is mediated by the quality of governance. Grigoli and Kapsoli [68] also emphasized the need to consider the efficiency of health spending. Simply increasing public expenditure in the health sector may not significantly affect health outcomes if the efficiency of spending is low. Therefore, it is crucial to ensure that health spending is allocated and utilized efficiently to achieve desired improvements in health outcomes. Similarly, the increased in temperature-related dengue risk may also be due to the higher health-seeking behavior of the population, especially in resource-rich locations. Consistent with a previous literature [69], hydrometeorological changes were found to substantially affect the risk of dengue in regions with mid-to-low latitudes in China. Patz et. al. [8] assert that increased incidence of dengue may first occur in regions bordering endemic zones in latitude or altitude. In summary, from a programmatic, local risk standpoint, our study shows that the association can be modified by area specific factors, thus merits the need for a tailor-fit dengue-control response. The Philippines may benefit from devolved decentralized and localized dengue control and early warning systems in order to account for these variabilities.

## Limitations

The current study has several limitations. Firstly, there is limited availability, accessibility, and data quality of surveillance data for dengue cases and meteorological variables for all provinces. The available dengue case data set is constrained by varied case definitions and under-reported dengue cases, as only clinically apparent cases were recorded. Locations with higher poverty incidence may be less likely to seek healthcare assistance due to less knowledge, accessibility, or financial capacity, resulting in lower reported dengue cases, and thus, inaccurately minimizing the modifying effect of poverty on the temperature-dengue relationship. These data sets were also subjected to varied implementation measures per province, making it difficult to determine if increased cases were related to improved surveillance or increases in

regional or global temperatures. Secondly, the absence of a historical time series of socio-economic variables has led us to treat data from 2015 as a representation of the data from 2010–2019. There is also a lack of complete, provincial data for other possibly modifying variables, such as forest cover and the presence of vegetation, open water containers, and degree of intra-household crowding. Future studies may benefit and have an improved understanding of dengue modeling if these additional spatial and socio-economic factors are included. Relationships between components of the model and the chosen socio-economic variables may change over time in ways that are difficult to predict, highlighting the need for more granular, long-term surveillance data.

## Conclusions

This observational study found that temperature was associated with dengue incidence, and that this association is more apparent in locations with high population density, less people in urban areas with larger household size, high poverty incidence, higher health spending per capita, and in lower latitudes. Differences with socio-economic conditions is linked with dengue risk. This highlights the need to develop interventions tailor-fit to local conditions.

## Consent for publication

Not applicable.

## Supporting information

**S1 Table. Sources of data meta-regressors.**
(DOCX)

**S2 Table. Description of socio-economic factors used for the study. The socio-economic factors used were province specific. All were originally yearly data, except for latitude.** The corresponding median values of the socio-economic factors were used, depending on the distribution of each dataset.
(DOCX)

**S3 Table. Main effects of the effect modifiers.**
(DOCX)

**S1 Fig. Correlation plot between the temperature data from ERA5-land and existing background monitoring stations in cities of Manila and Cebu.**
(DOCX)

**S2 Fig. Maximum lag sensitivity analyses.**
(DOCX)

**S3 Fig. Sensitivity analysis used for df selection.**
(DOCX)

**S4 Fig. Variation in the annual distribution of cases per year.**
(DOCX)

**S5 Fig. Sensitivity analyses adjusting for precipitation derived from ERA5-land.**
(DOCX)

**S6 Fig. Temperature-specific associations in different temperature percentiles.**
(DOCX)

**S7 Fig. Interaction analyses per effect modifier.**
(DOCX)

## Acknowledgments

We acknowledge the input from the Philippine Atmospheric, Geophysical, and Astronomical Services Administration (PAGASA) and data from the Philippine Statistics Authority (PSA).

## Author Contributions

**Conceptualization:** Xerxes Seposo, Sary Valenzuela, Geminn Louis Apostol.

**Data curation:** Xerxes Seposo, Sary Valenzuela, Geminn Louis Apostol.

**Formal analysis:** Xerxes Seposo.

**Funding acquisition:** Xerxes Seposo, Sary Valenzuela, Geminn Louis Apostol.

**Investigation:** Xerxes Seposo.

**Methodology:** Xerxes Seposo.

**Project administration:** Xerxes Seposo, Sary Valenzuela, Geminn Louis Apostol.

**Resources:** Sary Valenzuela, Geminn Louis Apostol.

**Software:** Xerxes Seposo.

**Visualization:** Xerxes Seposo.

**Writing – original draft:** Xerxes Seposo, Sary Valenzuela, Geminn Louis Apostol.

**Writing – review & editing:** Xerxes Seposo, Sary Valenzuela, Geminn Louis Apostol.

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
