## [Decision Letter · Decision Letter 0]

30 May 2023

Dear Dr. Seposo,

Thank you very much for submitting your manuscript "Socio-economic factors and its influence on the association between temperature and dengue incidence in 61 Provinces of the Philippines, 2010-2019" for consideration at PLOS Neglected Tropical Diseases. As with all papers reviewed by the journal, your manuscript was reviewed by members of the editorial board and by several independent reviewers. In light of the reviews (below this email), we would like to invite the resubmission of a significantly-revised version that takes into account the reviewers' comments. 

The reviewers raised concerns about the descriptions of the data, definitions of covariates, theoretical background for using the 18-week maximum lag, visualizations of the dlnm outputs, and the large uncertainty for the estimated effects in Figure 5.

We cannot make any decision about publication until we have seen the revised manuscript and your response to the reviewers' comments. Your revised manuscript is also likely to be sent to reviewers for further evaluation.

Sincerely,

Qu Cheng, Ph.D.

Guest Editor

Justin Remais

Section Editor

The reviewers raised concerns about the descriptions of the data, definitions of covariates, theoretical background for using the 18-week maximum lag, visualizations of the dlnm outputs, and the large uncertainty for the estimated effects in Figure 5.

Reviewer's Responses to Questions

**Key Review Criteria Required for Acceptance?**

**Methods**

-Are the objectives of the study clearly articulated with a clear testable hypothesis stated?

-Is the study design appropriate to address the stated objectives?

-Is the population clearly described and appropriate for the hypothesis being tested?

-Is the sample size sufficient to ensure adequate power to address the hypothesis being tested?

-Were correct statistical analysis used to support conclusions?

-Are there concerns about ethical or regulatory requirements being met?

Reviewer #1: I document major concerns in the attached. I am concerned about the omission of precipitation in the model, and in the parameterization of certain socioeconomic variables (e.g., is urban population used or percent urban population used?)

Reviewer #2: 1. The objectives of the study clearly articulated with a clear testable hypothesis stated - 1. to examine the association between dengue and temperature in multiple locations in the Philippines, and 2. to explore the modification role by socioeconomic factors.

2. The study design is appropriate to address the stated objectives - a modelling analysis has been carried out.

3. The population is clearly described and appropriate for the hypothesis being tested.

4. Correct statistical analysis were used to support conclusions - meta-regression is common to be used to summarize the effect size of the association.

5. There are concerns about ethical or regulatory requirements being met - No ethical statement has been mentioned.

Reviewer #3: Yes to all these questions.

**Results**

-Does the analysis presented match the analysis plan?

-Are the results clearly and completely presented?

-Are the figures (Tables, Images) of sufficient quality for clarity?

Reviewer #1: I document major concerns in the attached. I am concerned that the authors do not specific over which lags the exposure-response dimensions are calculated, nor for what temperature comparison the lag response dimension is calculated. I am also concerned about why the temperature-response relationship is highly uncertain for one SES axes of comparison, but not the other

Reviewer #2: Yes

Reviewer #3: I have a question regarding Figure 3 and the description about it in the manuscript. I raised this question in my reviewer attachment.

**Conclusions**

-Are the conclusions supported by the data presented?

-Are the limitations of analysis clearly described?

-Do the authors discuss how these data can be helpful to advance our understanding of the topic under study?

-Is public health relevance addressed?

Reviewer #1: I document major concerns in the attached. The authors should take care not to confuse interpretation of effect modification with main effects. Such is the case, for instance, in poverty incidence.

Reviewer #2: Yes

Reviewer #3: Yes to all.

**Editorial and Data Presentation Modifications?**

Reviewer #1: (No Response)

Reviewer #2: (No Response)

Reviewer #3: I recommend minor revision.

**Summary and General Comments**

Reviewer #1: In this study, authors examine how various socio-economic factors (including poverty, urbanization, and health-care expenditures) modify the temperature-dengue incidence across provinces in Philippines. They do so using a two-stage distributed lag non-linear model with meta-analysis. This research question is of importance, and the DLNM model is a good approach for this question. However, I have some methodological questions as well as some concerns with the presentation and interpretation of the results.

Reviewer #2: This is a modelling study 1. to examine the association between dengue and temperature in multiple locations in the Philippines, and 2. to explore the modification role by socioeconomic factors. Some comments:

1. According to the journal policy, the data are required to be shared to ensure a reproducibility.

2. The study findings are highly consistent with some others in South Asian settings that have not been cited e.g. doi.org/10.1016/j.jiph.2017.12.006, doi.org/10.1016/j.envint.2022.107518. Please discuss.

3. Please describe what is "ERA5-land"? Is the data source reliable?

4. What is the resolution of the socioeconomic factors?

5. Maximum lag of 18 weeks is too long even lag plot may support the assumption. Justification should be given.

6. A symbol cannpt be showed in line 129.

7. The first paragraph of results should report the incidence rate rather than number of cases.

8. The major concern of the study is that the uncertainty of categories is very large in the socioeconomic factors (Figure 5), so the results of the effect modification is very inconclusive. Even household size i don't think it is a effect modifier given both groups show a similar pattern. The authors should think of other ways to examine the modification effect.

Reviewer #3: (No Response)

PLOS authors have the option to publish the peer review history of their article (what does this mean?). If published, this will include your full peer review and any attached files.

Reviewer #1: No

Reviewer #2: No

Reviewer #3: No
---

## [Decision Letter · Decision Letter 1]

14 Sep 2023

Dear Dr. Seposo,

Thank you very much for submitting your manuscript "Socio-economic factors and its influence on the association between temperature and dengue incidence in 61 Provinces of the Philippines, 2010-2019" for consideration at PLOS Neglected Tropical Diseases. As with all papers reviewed by the journal, your manuscript was reviewed by members of the editorial board and by several independent reviewers. The reviewers appreciated the attention to an important topic. Based on the reviews, we are likely to accept this manuscript for publication, providing that you modify the manuscript according to the review recommendations. 

Sincerely,

Qu Cheng, Ph.D.

Guest Editor

Justin Remais

Section Editor

The reviewers still have minor suggestions regarding the visualizations, descriptions of the methods, and the interpretations of the results.

Reviewer's Responses to Questions

**Key Review Criteria Required for Acceptance?**

**Methods**

-Are the objectives of the study clearly articulated with a clear testable hypothesis stated?

-Is the study design appropriate to address the stated objectives?

-Is the population clearly described and appropriate for the hypothesis being tested?

-Is the sample size sufficient to ensure adequate power to address the hypothesis being tested?

-Were correct statistical analysis used to support conclusions?

-Are there concerns about ethical or regulatory requirements being met?

Reviewer #1: The objectives are clear (to describe the association between temperature and dengue incidence in the Philippines and to examine effect modification by SES variables) and the study design and DLNM is appropriate. No ethical concerns.

Reviewer #2: Yes to all

Reviewer #3: Yes to all.

**Results**

-Does the analysis presented match the analysis plan?

-Are the results clearly and completely presented?

-Are the figures (Tables, Images) of sufficient quality for clarity?

Reviewer #1: Generally yes. Please see comments for a few points raised, including about the axis of Figure 4, the need to state a reference level for Figure 4, and concerns about the model fit for the model using poverty incidence in the interaction variable.

Reviewer #2: Yes to all

Reviewer #3: Yes to all.

**Conclusions**

-Are the conclusions supported by the data presented?

-Are the limitations of analysis clearly described?

-Do the authors discuss how these data can be helpful to advance our understanding of the topic under study?

-Is public health relevance addressed?

Reviewer #1: Generally, yes. Please see comments for a few points raised about interpretation of main effects vs. interaction and interpretation of the model fit using poverty incidence in the interaction term.

Reviewer #2: Yes to all

Reviewer #3: Yes to all.

**Editorial and Data Presentation Modifications?**

Reviewer #1: (No Response)

Reviewer #2: Accept

Reviewer #3: Minor revision

**Summary and General Comments**

Reviewer #1: (No Response)

Reviewer #2: The authors have addressed my comments well.

Reviewer #3: Thank you for addressing my comments. I still have one question regarding the revised effect modification analysis. The authors stated that they centered the effect modifier by the 5th and 95th percentile values and ran the model (eq 2). Does it mean that the authors ran sperate models, one with effect modifier centered by the 5th percentile and the other with effect modifier centered by the 95th percentile? Or there is just one model with the effect modifier transformed to categorical values (0:<5th percentile, 1: 5-95th, 2:>95th) ? I think this part needs some further illustration so it is better understood by the general audience.

PLOS authors have the option to publish the peer review history of their article (what does this mean?). If published, this will include your full peer review and any attached files.

Reviewer #1: No

Reviewer #2: No

Reviewer #3: No

Figure Files:

Data Requirements:

Reproducibility:

References

---

## [Editor Report · Decision Letter 2]

4 Oct 2023

Dear Dr. Seposo,

We are pleased to inform you that your manuscript 'Socio-economic factors and its influence on the association between temperature and dengue incidence in 61 Provinces of the Philippines, 2010-2019' has been provisionally accepted for publication in PLOS Neglected Tropical Diseases.

Best regards,

Qu Cheng, Ph.D.

Guest Editor

Justin Remais

Section Editor

---

## [Editor Report · Acceptance letter]

19 Oct 2023

Dear Dr. Seposo,

We are delighted to inform you that your manuscript, "Socio-economic factors and its influence on the association between temperature and dengue incidence in 61 Provinces of the Philippines, 2010-2019," has been formally accepted for publication in PLOS Neglected Tropical Diseases.

Best regards,

Shaden Kamhawi

co-Editor-in-Chief

Paul Brindley

co-Editor-in-Chief
